# Fixed Orthodontic Treatment Increases Cariogenicity and Virulence Gene Expression in Dental Biofilm

**DOI:** 10.3390/jcm11195860

**Published:** 2022-10-03

**Authors:** Watcharawee Thanetchaloempong, Sittichai Koontongkaew, Kusumawadee Utispan

**Affiliations:** 1International College of Dentistry, Walailak University, Bangkok 10300, Thailand; 2Faculty of Dentistry, Thammasat University, Pathum Thani 12120, Thailand

**Keywords:** brackets, cariogenicity, dental biofilm, dental caries, fixed orthodontic appliances, oral health, virulence genes

## Abstract

Background: Dental caries commonly occurs during orthodontic treatment because fixed appliances can impede effective oral hygiene practices. This study investigated the effects of fixed orthodontic treatment on dental biofilm maturity and virulence gene (*gtfB*, *ldh*, *brpA*, *spaP*, *luxS*, and *gbpB*) expression. Methods: Dental biofilms and virulence gene expression were determined in 24 orthodontic patients before and after treatment of ≥6 months. A three-tone disclosing gel was used to stain dental biofilm and assess its maturity by its color change—pink (new dental biofilm), purple (mature dental biofilm), and light blue (cariogenic dental biofilm). Gene expression levels were determined using real-time PCR. Results: After fixed orthodontic appliance insertion, the percentage of new dental biofilm decreased, whereas that of cariogenic dental biofilm significantly increased (*p* < 0.05). There was no significant difference in the percentage of mature dental biofilm (*p* > 0.05). Fixed orthodontic appliances increased *gtfB*, *ldh*, *brpA*, and *gbpB* gene expression above 1.5-fold in dental biofilm. In contrast, there was no change in *spaP* or *luxS* gene expression after treatment. Conclusions: Fixed orthodontic appliance insertion induced ecological changes and cariogenic virulence gene expression in dental biofilm.

## 1. Introduction

Dental caries is one of the most common public health problems and is induced by multiple risk factors involving dental biofilm (also known as dental plaque), diet habits, oral hygiene, socioeconomic status, and related genetic disorders such as molar/incisor hypomineralization [1]. Dental caries occurs due to a substrate disruption of the bacterial community, as described in the “ecological plaque hypothesis”. The ecological plaque hypothesis proposes that any major changes in local environmental conditions, e.g., high sucrose consumption, induces the colonization of cariogenic bacteria in the dental biofilm and results in cariogenic dental biofilm formation [2]. In addition to focusing on specific bacteria, the bacterial virulence/function and dental biofilm ecology should be considered when assessing the cariogenicity of dental biofilm [3,4].

Dental caries is commonly observed in patients undergoing orthodontic treatment [5]. Fixed orthodontic appliances, including brackets, springs, and arch wires, impede access to the tooth surface, making it difficult to remove dental biofilm by mechanical cleaning [6,7]. A systematic review revealed that the presence of fixed appliances influences the quantity and quality of the oral microbiota. These appliances create an ecological environment favorable to qualitative and quantitative changes in dental biofilm microorganisms [8]. The number of cariogenic bacteria, including *Streptococcus mutans* and lactobacilli, increases in the dental biofilm on teeth with fixed orthodontic appliances [9].

*S. mutans* is an oral bacterium belonging to the mutans streptococci (MS) group. It is an important bacterium that participates in dental biofilm formation and dental caries. Virulence genes in *S. mutans* can be classified into genes involved in bacterial adhesion, extracellular polysaccharide formation, dental biofilm formation, sugar uptake and metabolism, acid tolerance, and biofilm regulation mechanisms [10]. The genes involved in bacterial adhesion are *gbp*s (*gbpA*, *gbpB*, and *gbpC*) and *spaP*. The *gbp* genes encode glucan-binding proteins (GBP) [11]. The *spaP* gene encodes the cell surface antigen (SpaP), which is the adhesion protein mediating early bacterial attachment in dental biofilm [12]. *S. mutans* contains *gtfB*, *gtfC*, and *gtfD* genes, each of which encodes glucosyltransferases (GTFs). GTFs convert sucrose to glucans. Glucans mediate sucrose-dependent adherence of *S. mutans* to the tooth surface, and act as a component of the dental biofilm matrix [13]. In *S. mutans*, *ldh* encodes lactate dehydrogenase (LDH), which is involved in lactic acid production [14]. The *brpA* gene regulates *S. mutans* cell envelope integrity [15], biofilm formation, and acid tolerance [16,17]. The *luxS* gene regulates quorum sensing within dental biofilm and regulates cellular metabolism, DNA repair, and stress tolerance in *S. mutans* [18,19].

Various methods have been used to assess caries risk, including dental biofilm staining, bacterial culture, and molecular biology techniques, such as polymerase chain reaction (PCR). Among these methods, PCR has been preferentially used to detect cariogenic bacteria and/or putative MS virulence genes because it is quick, sensitive, and simple [10,20].

Assessing dental biofilm is difficult for patients and dentists because the tooth and dental plaque often appear similar, especially if dental biofilm is present in small amounts. The dental biofilm is typically detected by clinicians, either by directly using an explorer or with a disclosing solution, and is quantified using indices based on the area of the tooth covered or its thickness [21,22]. Furthermore, most orthodontic trials have used plaque indices that scored the extent and thickness of dental biofilm as ordinal scales [23,24]. Therefore, in previous studies, the usefulness of a dental biofilm index for patients with fixed orthodontic appliances depended on the subjective evaluation of the extent of stained dental biofilm by the dentist. Furthermore, the disclosing agents previously used to stain mature and newly formed dental biofilm lack specificity. These assessment methods have a limitation of being subjective; therefore, the results may vary between clinicians, especially when little dental biofilm is present. Thus, there is a need to more precisely evaluate the dental biofilm in orthodontic patients [23,25]. A three-tone plaque disclosing gel (GC Tri Plaque ID Gel^TM^) was developed to identify the caries risk in individuals [26]. This gel contains red and blue pigments, and sucrose. The new dental biofilm is stained pink/red because its structure is sparse, and the blue pigment easily washes off. In contrast, the mature dental biofilm (>48 h) structure is dense; thus, the blue and red pigments are trapped, resulting in blue/purple staining. In the acid-producing mature dental biofilm, the sucrose in the disclosing gel is metabolized by acidogenic bacteria in the dental biofilm. When the dental biofilm pH drops to <pH 4.5, the dental biofilm becomes light blue. In addition, the light blue-stained dental biofilm has been found to have high levels of *S. mutans* [27]. Therefore, the light blue dental biofilm is cariogenic dental biofilm, which is the most virulent, followed by the blue/purple, and pink dental biofilm.

Currently, there is no evidence supporting the use of three-tone plaque disclosing gel to assess the cariogenic status of dental biofilm in orthodontic patients. The present study used a three-tone disclosing agent and investigated the virulence-related gene expression by real-time PCR, to determine dental biofilm cariogenicity in fixed orthodontic patients. In this study, we tested the hypothesis that fixed orthodontic appliances affect the cariogenicity (primary outcome) and virulence of dental biofilm (secondary outcome). Therefore, the aim of the present study was to investigate the effect of orthodontic treatment on dental biofilm maturity and virulence gene expression levels, using a three-tone plaque disclosing gel and real-time PCR, respectively. The association between the virulence genes of cariogenic bacteria and the cariogenicity of dental biofilm was also evaluated.

## 2. Materials and Methods

### 2.1. Participants and Background

This study received ethical approval from the Human Research Ethics Committee of Walailak University, Thailand (no. WUEC-20-225-01/2), on 10 August 2020. The principles of the Declaration of Helsinki were followed in the present study [28]. This study was a nonrandomized open label trial (Figure 1). The patients were ≥ 12-year-old individuals, who were undergoing fixed orthodontic treatment at the Center for Advanced Oral Health, Walailak University. The exclusion criteria were having a systemic disease or being on a long-term/recent/current regimen of medications affecting salivary flow, receiving antibiotic therapy within the previous 6 months, and/or severe periodontal problems. Written consent was obtained from the participants or a parent of the participants under 16 years old on the first day of the study. The participants were recruited using a purposive sampling method, based on the inclusion and exclusion criteria, which resulted in 24 subjects.

### 2.2. Sample Size Calculation

The sample size calculation was performed to detect the effect of orthodontic treatments on the primary outcome (dental biofilm maturity) [29,30] using G*power 3.1.9.4. (Heinrich-Heine University, Dusseldorf, Germany). Because there were no exact effect size values in previous reports, an estimation of the effect size was computed using Cohen’s d to obtain the standardized effect size [31]. The paired *t*-test was used to detect the effects of fixed orthodontic treatment on dental biofilm maturity (power of 0.8, α error of 0.05, and Cohen’s d of 0.8). This calculation determined that a minimum of 15 patients were required. To compensate for dropouts, 24 patients were enrolled in this study.

### 2.3. Questionnaire and Food Diary

Interview questionnaires were used to obtain information about the patients’ demographic characteristics, orthodontic treatment duration, and oral hygiene practice. A 3-day food diary was distributed to each patient. The patients were required to record their food intake in the food diary on two weekdays and one weekend day [32]. In the present study, the patients were educated and motivated to enter everything that he/she consumed from morning until bedtime in the chart. They were also requested to include any medications, chewing gum, and cough lozenges. The chart was analyzed after orthodontic treatment for at least 6 months by the dentist–patient duo for the frequency of acid foods, including frank or occult sugar intake between meals. The average number of exposures to sugar and acid between meals over the 3 d was calculated for each patient [33]. In addition, the patients were educated on essential oral hygiene maintenance at home during orthodontic treatment. The patients were instructed to regularly brush their teeth with the modified Bass technique, using a manual or powered toothbrush with fluoride toothpaste, for at least 2 min after every meal. An interdental brush was recommended to clean the small spaces under the wires and around the bands and brackets.

### 2.4. Dental Biofilm Staining

At the beginning of the study, dental biofilm staining and collection were performed before orthodontic treatment. The dental biofilm was stained and collected again after treatment for at least six months. All tooth surfaces were stained, except for the occlusal surface, as previously described [34,35]. The dental biofilm maturity was assessed using GC Tri Plaque ID Gel ^TM^ (GC Corporation, Tokyo, Japan), according to the manufacturer’s protocol. Briefly, the gel was applied with a microbrush on all tooth surfaces and left undisturbed for 2 min. The tooth surfaces were then gently rinsed with water for 30 sec, and the changes in dental biofilm color were observed (Figure 2). The original dark blue changed to light blue when the pH in dental biofilm was less than 5. Immature and mature dental biofilm stained pink and purple, respectively. The most mature dental biofilm was recorded for each tooth. Based on the color changes on the tooth surfaces, the dental biofilm (plaque) maturing staining (PMS) was obtained using the formula:% PMS = (number of teeth with each plaque color/total number of teeth examined) × 100
as previously described [35].

### 2.5. Dental Biofilm Collection

Dental biofilm samples were collected from the tooth surfaces with a sterile dental explorer. Each sample was placed in a sterile 1.5 mL microcentrifuge tube. The sample was kept on dry ice and transported to the laboratory and stored at −80 °C until use.

### 2.6. Determination of Gene Expression by Real-Time PCR

Total RNA of the dental biofilm samples was extracted using Trizol reagent (Invitrogen, Carlsbad, CA, USA). Complementary DNA was synthesized from total RNA (1 µg) using a PrimeScript 1st strand cDNA Synthesis Kit (Takara Bio Inc., Shiga, Japan). Real-time PCR was performed using a KAPA SYBR^®^ FAST qPCR Kit Master Mix (Kapa Biosystems, Wilmington, MA, USA) in a QuantStudio™ 3 Real-Time PCR System (Thermo Fisher Scientific, Waltham, MA, USA). The PCR primer sequences were obtained from previous studies (Table 1). The *gtfB, ldh, brpA, spaP, luxS,* and *gbpB* primers were obtained from Wen et al. [36]. The *16S rRNA* gene served as the internal control [37]. The relative gene expression levels were determined using the 2^−ΔΔCT^ equation. In this case, ΔCT = CT (After treatment group)—CT (Before treatment group). In this study, we defined Differentially Expressed Genes (DEGs) based on the fold changes in gene expression between before and after orthodontic treatments. The DEGs were considered if their expression demonstrated at least a 1.5-fold increase [38].

### 2.7. Statistical Analysis

The statistical analysis was performed using GraphPad Prism Version 7.0 (GraphPad software Inc., La Jolla, CA, USA) and the Statistical Package of Social Sciences (SPSS) version 25 (IBM Corp., Armonk, NY, USA). Descriptive statistics are displayed as the mean, standard deviation (SD), and median and range for quantitative variables where appropriate. Frequencies and percentages were used to describe qualitative data. Numerical data were plotted in the figures as the mean ± the standard error of the mean (SEM) and analyzed for statistical significance using the paired *t*-test. Multiple linear regression analysis was used to predict the association of the outcome variable (the PMS of the cariogenic dental biofilm) with the predictor variables (the expression level of virulence genes). Multicollinearity was assessed using the variance inflation factor (VIF) and tolerance values. A variance inflation factor (VIF) > 10 and/or intolerance < 0.1 indicates serious multicollinearity [39]. The significance level for all tests was defined as *p*-value ≤ 0.05.

## 3. Results

### 3.1. The Study Population

The patients’ demographic characteristics, treatment durations, oral hygiene practices, and dietary habits are shown in Table 2. The study comprised 24 patients, 14–41 years old (mean ± SD = 27.21 ± 6.45). There were 12 (50%) males and 12 (50%) females. Dental biofilm staining and determination of virulence gene expression were performed after orthodontic treatment for 7.74 ± 0.65 months (mean ± SD). All patients used fluoride toothpaste. The brushing frequencies of most patients were ≥2 times/day. They occasionally used dental floss, an interdental brush, and mouthwash. The patients consumed sugary and acidic food between meals 0.9 (0–3) and 0 (0–0.4) (median and range) times per day, respectively.

### 3.2. Dental Biofilm Maturity

The percent PMS of new dental biofilm significantly decreased after fixed orthodontic treatment (*p* < 0.05) (Figure 3). In contrast, the cariogenic dental biofilm percentage significantly increased after treatment (*p* < 0.05). However, there was no significant difference in the PMS of mature dental biofilm before and after treatment (*p* > 0.05).

### 3.3. Cariogenic Virulence Gene Expression in Dental Biofilm

Representative cariogenic genes (*gtfB*, *ldh*, *brpA*, *spaP*, *luxS*, and *gbpB*) in *S. mutans* were investigated in dental biofilm using real-time PCR. After fixed orthodontic treatment, DEGs were found in *gtfB*, *ldh*, *brpA*, and *gbpB* (Figure 4). The expression of *spaP* or *luxS* following the insertion of fixed orthodontic appliances was similar to that pre-insertion.

### 3.4. Association between PMS and Virulence Gene Expression Level

Multiple linear regression analyses were used to predict the association between the PMS of cariogenic dental biofilm and virulence gene expression levels. In this study, the outcome and predictor variables were the PMS and *gtfB*, *ldh*, *brpA*, *spaP*, *luxS*, and *gbpB* expression levels, respectively. The results revealed no significant correlation between cariogenicity and the expression level of virulence genes in dental biofilm after fixed orthodontic treatment (*p* > 0.05, R^2^ = 0.25) (Table 3). In addition, the predictor’s tolerance values were greater than 0.1 and the VIF values were less than 10. Therefore, there was no significance of multicollinearity among our predictor variables.

## 4. Discussion

Dental caries is a multifactorial dynamic disease; the presence of dental biofilm and its interaction between host factors and sugar is very important in caries development [40]. This interaction influences the dental biofilm environment, as explained by the ecological plaque hypothesis, and acidogenic bacteria play an important role in dental biofilm acidogenicity [40,41]. A previous study determined the association between acidogenic bacterial counts and dental biofilm maturity using three-tone plaque disclosing gels [42]. A high amount of acidogenic bacteria was found in light blue dental biofilm, followed by purple and pink. The present study found that the PMS of cariogenic dental biofilm was significantly (18.54%) higher in the orthodontic patients after fixed appliance insertion. Therefore, our results confirm those of previous reports.

The presence of fixed orthodontic appliances makes oral hygiene more challenging for orthodontic patients. Fixed appliances impede access to the tooth surface and increase the complexity of the mechanical cleaning that the patient must perform during orthodontic treatment. Poor oral hygiene practices, such as ineffective tooth brushing, and dental biofilm accumulation on orthodontic appliances, can drive ecological changes in dental biofilm around the appliances. Multicolor plaque disclosing dyes allow clinicians to rapidly identify where patients are struggling with mechanical cleaning. This can be very informative for identifying high caries risk areas around brackets that can guide dental professionals to efficiently provide targeted oral hygiene education [26,27,34].

In this study, *gtfB*, *gbpB*, *ldh*, and *brpA* expression levels and dental biofilm cariogenicity in orthodontic patients increased after fixed orthodontic treatment. These findings are partially in line with our previous study in asthmatic patients [35]. However, in addition to *gtfB*, *gbpB*, *ldh*, and *brpA*, *spaP*
*and*
*luxS* gene expression was significantly increased in asthmatic patients. It is possible that a relatively high caries risk status is common in asthmatic patients. A meta-analysis found that the caries risk level was double in asthmatic patients [43]. The most asthmatic patients have been found to have a significant decrease in salivary flow rates, pH, and buffer capacity, which are known to predispose individuals to dental caries.

The *ldh* gene in *S. mutans* has been reported to be constitutively expressed [36]. During the sugar fermentation process, LDH catalyzes the transformation of pyruvate (plus NADH) to lactic acid plus NAD^+^ at the final step. This reaction is exergonic, thermodynamically preferred, but reversible, and lactate can be oxidized to pyruvate and NADH, if needed. Most oral acidogenic bacteria produce LDH by activating *ldh* gene expression, which directly correlates with their acid production and cariogenicity [44]. We therefore hypothesize that *ldh* expression could be a suitable biomarker to correlate the relative metabolic activity with the acid production within cariogenic dental biofilm. Indeed, 5.5-fold increases in *ldh* expression and cariogenic dental biofilm were observed following orthodontic treatment.

Similar results were also observed in the expression of *gtfB*, *gbpB,* and *brpA*. GTFs and GBP of *S. mutans* are known to be differentially expressed in response to environmental conditions, such as carbohydrate source and availability, pH, and growth of the bacteria on surfaces [10]. GTFs and GBP facilitate bacterial adherence and biofilm accumulation [11,13,45]. In the present study, *gtfB* and *gbpB* demonstrated increased expression after orthodontic treatment, whereas there was no change in *spaP* expression in the cariogenic dental biofilm of our patients. The *spaP* gene is considered the primary factor mediating the early attachment of *S. mutans* to tooth enamel in the absence of sucrose [46]. In acidic dental biofilm, which had the highest %PMS in our patients, *spaP* might not be predominantly expressed compared with *gtfB* and *gbpB*. The expression of *brpA* in *S. mutans* has been proposed to be associated with acid tolerance and biofilm development [15,17]. The increased *brpA* expression found in the present study might play an important role in dental biofilm cariogenicity after fixed orthodontic appliance insertion.

Although *luxS* expression in *S. mutans* is associated with biofilm formation and biofilm structure maintenance, its roles in regulating the factors critical to biofilm formation are unresolved [36]. However, it has been found that the LuxS quorum sensing system is considered the main system that most of oral bacteria use to communicate within biofilm [47]. The most common and widespread quorum sensing signal among bacteria is autoinducer-2, which is a signaling molecule induced by LuxS that regulates the expression of numerous genes, including *gtfB/C* genes [48]. Our results suggest that *luxS* gene expression might upregulate *gtfB*. In addition, *luxS* might not be involved in the cariogenic environment induced in dental biofilm after fixed orthodontic treatment. These results imply that the expression level of virulence genes and their related factors may be altered in dental biofilm, when the patients have fixed orthodontic appliances inserted.

We also determined the influence of virulence genes on the cariogenicity of dental biofilm. Multiple linear regression analysis indicated that there was no significant correlation between cariogenicity and virulence gene expression in dental biofilm after fixed orthodontic treatment. Our results suggest that the high cariogenicity of dental biofilm, formed in the presence of fixed orthodontic appliances, may be explained by the overexpression of the virulence genes involved in bacterial adherence and biofilm formation. However, the cariogenicity of dental biofilm cannot be totally explained by virulence gene activation. Other components/conditions in dental biofilm, such as a high concentration of insoluble glucans in its matrix, the low inorganic concentration, and its protein composition may contribute to dental biofilm cariogenicity [49,50].

Surprisingly, high sugary and acidic food consumption were not observed in our patients. A high frequency of sugary (>1 time/day) and acidic food (>2 times/day) intake between meals is considered high caries risk behavior [51]. Most of our patients were exposed to sugary and acidic food between meals at a frequency of less than 1 time/day. Therefore, sugary and acidic food consumption in our patients might not be factors involved in the increase in cariogenic dental biofilm after fixed orthodontic appliance insertion. These dietary history results did not agree with our previous findings in asthmatic patients, who were high caries risk individuals [35]. The average frequency of sugary food intake between meals in asthmatics was three times/day, which was higher than in the control subjects. The disparate results in orthodontic patients might be due to the impact of dietary counseling during orthodontic treatment.

Fixed orthodontic therapy may provide retentive site-associated dental biofilm accumulation [52]. Our findings imply that fixed orthodontic appliances may induce oral ecologic changes, which lead to increased dental biofilm cariogenicity. Increased expression of specific virulence genes related to dental biofilm formation, acid production, and acid tolerance contributed to greater cariogenicity. Moreover, acidogenicity and virulence gene expression in dental biofilm may be altered in response to cariogenic conditions, including orthodontic treatment and asthma. This study suggests that dental biofilm staining by a three-tone plaque disclosing gel is useful for assessing the cariogenic dental biofilm in orthodontic patients. Using a three-tone plaque disclosing gel for assessing dental biofilm in combination with the application of a biomimetic hydroxyapatite product may improve the efficiency of caries prevention in orthodontic patients, by reducing the incidence of white spot lesions along the surface of the brackets [53]. However, our results should be interpreted with caution because the level of acidogenic bacteria, particularly *S. mutans* in the dental biofilm, was not investigated in this study.

## 5. Conclusions

This study’s results indicate that staining dental biofilm with a three-tone disclosing gel allows for cariogenic dental biofilm assessment, following orthodontic treatment. Real-time PCR analysis demonstrated that the expression of virulence genes related to biofilm formation (*gtfB* and *gbpB*), acid production (*ldh*), and acid tolerance (*brpA*) were altered in response to fixed orthodontic appliance insertion. Our results suggest that cariogenicity and virulence gene expression in dental biofilm can be used in the early identification of caries risk status in orthodontic patients. A three-tone plaque disclosing dye can allow orthodontists to quickly identify the tooth surfaces where patients are struggling with mechanical cleaning. In combination with good oral hygiene maintenance, assessing cariogenicity and virulence gene expression in dental biofilm may synergistically reduce the incidence of carious lesions in orthodontic patients.

## Figures and Tables

**Figure 1 jcm-11-05860-f001:**
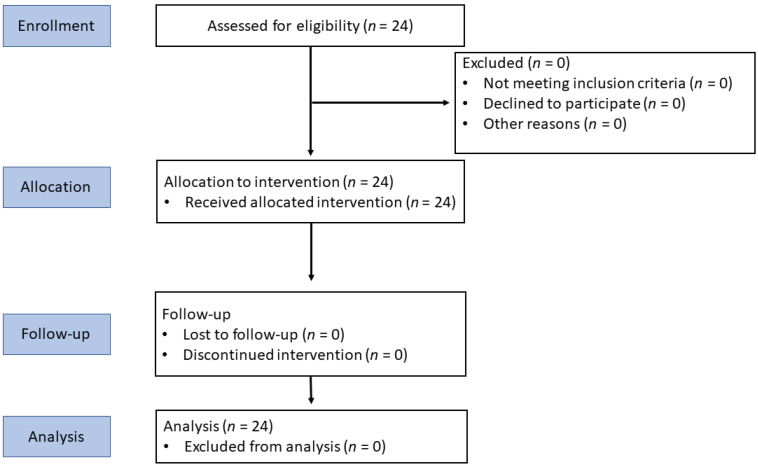
Flow diagram. Twenty-four patients completed the full study protocol.

**Figure 2 jcm-11-05860-f002:**
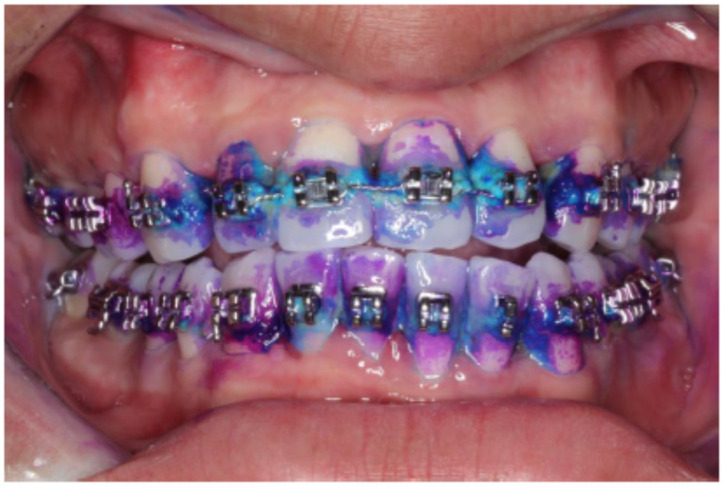
Use of the three-tone plaque disclosing gel, showing cariogenic, mature, and new dental biofilm as light blue, dark purple/blue, and pink/red stained areas, respectively.

**Figure 3 jcm-11-05860-f003:**
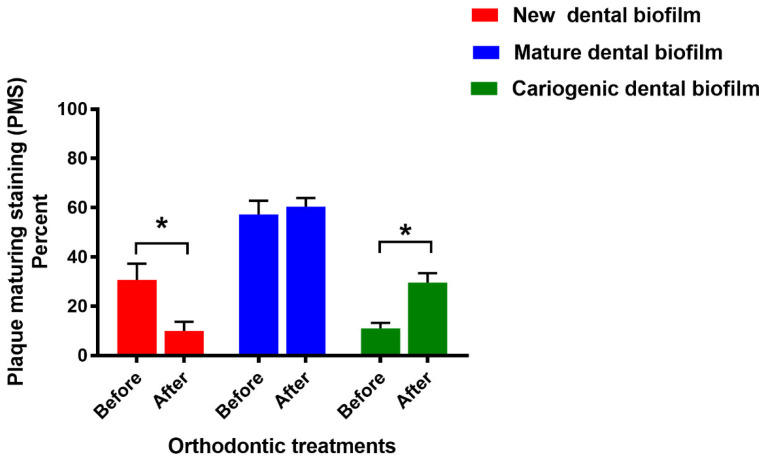
Percent PMS in the patients before and after fixed orthodontic treatment (*n* = 24). Bars represent the PMS mean ± SEM. Asterisks indicate a significant difference (*p* < 0.05) using the paired *t*-test.

**Figure 4 jcm-11-05860-f004:**
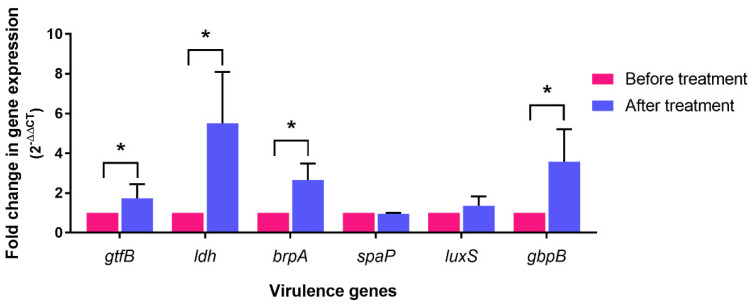
Fold change in *gtfB*, *ldh*, *brpA*, *spaP*, *luxS*, and *gbpB* gene expression in dental biofilm after orthodontic treatment (*n* = 24). Bars represent the mean ± SEM of gene expression level relative to that of the expression level before treatment. Asterisks indicate Differentially Expressed Genes (DEGs), which were defined as at least a 1.5-fold increase.

**Table 1 jcm-11-05860-t001:** Primer sequences for real-time PCR.

Primers	Forward Primers (5′–3′)	Reverse Primers (5′–3′)	Size (bp)
*gtfB*	AGCAATGCAGCCATCTACAAAT	ACGAACTTTGCCGTTATTGTCA	98
*ldh*	TTGGCGACGCTCTTGATCTTAG	GTCAGCATCCGCACAGTCTTC	92
*brpA*	CGTGAGGTCATCAGCAAGGTC	CGCTGTACCCCAAAAGTTTAGG	148
*spaP*	TCCGCTTATACAGGTCAAGTTG	GAGAAGCTACTGATAGAAGGGC	121
*luxS*	ACTGTTCCCCTTTTGGCTGTC	AACTTGCTTTGATGACTGTGGC	93
*gbpB*	CGTGTTTCGGCTATTCGTGAAG	TGCTGCTTGATTTTCTTGTTGC	108
*16S rRNA*	TCCACGCCGTAAACGATGA	TTGTGCGGCCCCCGT	119

**Table 2 jcm-11-05860-t002:** Demographic data, treatment duration, oral hygiene practices, and dietary habits of the orthodontic patients.

Variables	Values
Demographic characteristics	
Age (years) Mean ± SD Min–Max	27.21 ± 6.45 14–41
Sex Male, *n* (%) Female, *n* (%)	12 (50%) 12 (50%)
Orthodontic treatment duration	
Fixed orthodontic therapy follow-up (months) Mean ± SD Min–Max	7.74 ± 0.63 6.46–8.70
Oral hygiene practices	
Tooth brushing frequency, *n* (%) 1 time a day ≥2 times a day	1 (4.7) 23 (95.83)
Use of dental floss, *n* (%) Daily Occasionally No	4 (16.67) 15 (62.5) 5 (20.83)
Use an interdental brush, *n* (%) Daily Occasionally No	9 (37.5) 15 (62.5) 0 (0)
Use a fluoride toothpaste, *n* (%) Yes No	24 (100) 0 (0)
Use mouthwash, *n* (%) Yes No	13 (54.17) 11 (45.83)
Dietary habits	
Sugary intake between meals/day (Median and range)	0.90 (0–3)
Acidic food intake between meals/day (Median and rage)	0 (0–0.4)

**Table 3 jcm-11-05860-t003:** Analysis of the association between the PMS of cariogenic dental biofilm and the expression level of virulence genes using multiple linear regression.

Predictor Variables	Coefficients		*p*-Value	Collinearity Statistics	
	Unstandardized Coefficient (B)	Coefficients Standard Error		Tolerance	VIF
*gtfB*	−1.15	1.2	0.35	0.88	1.14
*ldh*	−0.19	0.44	0.67	0.49	2.06
*brpA*	−0.71	0.16	0.65	0.39	2.56
*spaP*	−16.37	23.1	0.49	0.98	1.02
*luxS*	−2.45	1.71	0.17	0.96	1.04
*gbpB*	0.14	0.59	0.82	0.68	1.46

## Data Availability

Not applicable.

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
