# Peer review of "Fixed Orthodontic Treatment Increases Cariogenicity and Virulence Gene Expression in Dental Biofilm"

_jcm, 2022, doi:10.3390/jcm11195860_

Round 1
Reviewer 1 Report
1. The effect size was reported in the Participants and Background not in the Sample Size Calculation, while the effect was not reported in the latter section and how did the author estimate it
2. The frequencies 126 of meals, sweets, sugary drinks, and acidic beverage intake were calculated as previously 127 described [32]. How? Needs more clarification
Author Response
Response to reviewer 1
We thank the Reviewer for the careful reading of our manuscript and giving suggestions that have allowed us to improve the revised version. The revised passages based on the reviewer’s comments are highlighted in yellow in the re-submitted manuscript. Our responses to the comments are found below:
- The effect size was reported in the Participants and Background not in the Sample Size Calculation, while the effect was not reported in the latter section and how did the author estimate it.
Response 1: We have removed the effect size from the Participants and Background section.
There was no effect size value from the previous reports. Therefore we used the standardized effect size (Cohen’s d = 0.8) to measure the mean differences in PMS between groups as previously suggested [30]. In our study, a power of 0.8, α error of 0.05 and standardized effect size Cohen’s d of 0.8, were used to estimate the minimum sample size. This estimation determined that a minimum of 15 patients were required, and 24 patients were enrolled to compensate for dropouts.
These contents were described in the Sample Size Calculation section (line 132-137)
- The frequencies of meals, sweets, sugary drinks, and acidic beverage intake were calculated as previously described [32]. How? Needs more clarification
Response 2: We added the dietary analysis method to clarify the food diary record and calculation (line 143-148).
Revised text: “In the present study, the patients were educated and motivated to enter everything that he/she consumed from morning until bedtime in the chart. They were also requested to include any medications, chewing gum, and cough lozenges. The chart was analyzed after orthodontic treatment for at least 6 months by the dentist-patient duo for the frequency of acid foods, including frank or occult sugar intake between meals.”
Reviewer 2 Report
Manuscript of considerable interest for specialists experienced in orthodontics and in dental hygiene, before proceeding to the publications need for revisions.
Remove the numbers in the abstract
Increase the keywords, the present are few.
Introduction, adding the various risk factors that can induce caries, but the onset of MIH is increasingly widespread, both from a pre-eruptive and post-eruptive and genetic point of view, have already been studied by Prof. Scribante's research group.
Materials and methods: in addition to food education what were the tips for home oral hygiene, nowadays an electric and / or sonic toothbrush is essential.
In Figure 2, better specify the time of each staining
The results are not easily decipherable by the reader, make them more usable and highlight the statistically significant data.
Discussion, add all the prophylaxis systems, the use of biomimetic hydroxypatite (also studied by the research group of Prof. Scribante) in order to reduce the incidence of white spots along the surface of the brackets, and formulate a proactive approach rather than reactive.
Conclusions, rephrase it by adding proactive action
Bibliography: add required references.
Author Response
Response to reviewer 2
We thank the Reviewer for the careful reading of our manuscript and giving suggestions that have allowed us to improve the revised version. The revised passages based on the reviewer’s comments are highlighted in yellow in the re-submitted manuscript. Our responses to the comments are found below:
Manuscript of considerable interest for specialists experienced in orthodontics and in dental hygiene, before proceeding to the publications need for revisions.
- Remove the numbers in the abstract
Response 1: The numbers in the abstract were removed.
- Increase the keywords, the present are few.
Response 2: The keywords were increased to 7 words.
Revised text:
“Keywords: brackets; cariogenicity; dental biofilm; dental caries; fixed orthodontic appliances; oral health; virulence genes”
- Introduction, adding the various risk factors that can induce caries, but the onset of MIH is increasingly widespread, both from a pre-eruptive and post-eruptive and genetic point of view, have already been studied by Prof. Scribante's research group.
Response 3: The factor contributing dental caries including genetic disorder; molar incisor hypomineralisation was added in the Introduction section (line 30-33).
Revised text: “Dental caries is one of the most common public health problems and is induced by multiple risk factors involving dental biofilm (also known as dental plaque), diet habits, oral hygiene, socioeconomic status, and related genetic disorders such as molar/incisor hypomineralisation [1].”
- Materials and methods: in addition to food education what were the tips for home oral hygiene, nowadays an electric and / or sonic toothbrush is essential.
Response 4: The essential oral hygiene maintenance tips for the orthodontic patients were added in the Questionnaire and Food Diary section of Materials and Methods part (line 149-154).
Revised text: “In addition, the patients were educated for essential oral hygiene maintenance at home during orthodontic treatment. The patients were instructed to regularly brush their teeth with the Modified Bass technique using a manual or powered toothbrush, with fluoride toothpaste for at least 2 min after every meal. An interdental brush was recommended to clean the small spaces under the wires and around the bands and brackets.”
- In Figure 2, better specify the time of each staining
Response 5: We apologize for our miscommunication of dental biofilm staining using the Tri-plaque ID gel. Basically, the gel appears as dark blue dye as original color before reacting to the dental biofilm. Once the gel was applied onto the tooth surfaces, its original color would be simultaneously changed into pink/purple/light blue, based on the quality of dental biofilm. Therefore, we revised added the brief method of staining in the Dental Biofilm staining section of Materials and Methods part (line 158-164).
Revised text: “The dental biofilm was stained and collected again after treatment for at least six months. All tooth surfaces were stained, except for the occlusal surface, as previously described [34, 35]. The dental biofilm maturity was assessed using GC Tri Plaque ID Gel TM (GC Corporation, Tokyo, Japan), according to the manufacturer’s protocol. Briefly, the gel was applied with a microbrush on all tooth surfaces and left undisturbed for 2 min. The tooth surfaces were then gently rinsed with water for 30 sec, and the changes in dental biofilm color were observed (Figure 2).”
- The results are not easily decipherable by the reader, make them more usable and highlight the statistically significant data.
Response 6: We revised some parts of results by emphasizing the significant data.
Revised text:
3.2 Dental Biofilm Maturity
“The percent PMS of new dental biofilm significantly decreased after fixed orthodontic treatment (p < 0.05) (Figure 3). In contrast, the cariogenic dental biofilm percentage significantly increased after treatment (p < 0.05). However, there was no significant difference in the PMS of mature dental biofilm before and after treatment (p > 0.05). (line 276-280).”
3.3 Cariogenic Virulence Gene Expression in Dental Biofilm
“After fixed orthodontic treatment, DEGs were found in gtfB, ldh, brpA, and gbpB (Figure 4). The expression of spaP or luxS following the insertion of fixed orthodontic appliances was similar to that pre-insertion (line 295-297).”
Figure 4.
“Asterisks indicate Differentially Expressed Genes (DEGs), which was defined as at least a 1.5-fold increase (line 305-306).”
- Discussion, add all the prophylaxis systems, the use of biomimetic hydroxypatite (also studied by the research group of Prof. Scribante) in order to reduce the incidence of white spots along the surface of the brackets, and formulate a proactive approach rather than reactive.
Response 7: We added the benefit of biomimetic hydroxyapatite use to prevent initial caries in orthodontic patients with the suggested reference in the Discussion part (line 418-421).
Revised text: “Using the three‑tone plaque disclosing gel for assessing dental biofilm in combination with the application of a biomimetic hydroxyapatite product may improve the efficiency of caries prevention in orthodontic patients by reducing the incidence of white spot lesions along the surface of the brackets [54].”
- Conclusions, rephrase it by adding proactive action
Response 8: We revised the Conclusion part by adding proactive action for dental caries prevention in the orthodontic patients (line 434-440).
Revised text: “Our results suggest that cariogenicity and virulence gene expression in dental biofilm can be used in the early identification of caries risk status in orthodontic patients. A three-tone plaque disclosing dye can allow orthodontists to quickly identify the tooth surfaces where patients are struggling with mechanical cleaning. In combination with good oral hygiene maintenance, assessing cariogenicity and virulence gene expression in dental biofilm may synergistically reduce the incidence of carious lesions in orthodontic patients.”
- Bibliography: add required references.
Response 9: The required references were added at no. [1] and [54].
Round 2
Reviewer 2 Report
the manuscript has been correctly revised, it can be published